# How to Determine the Most Powerful Pre-trained Language Model without Brute Force Fine-tuning? An Empirical Survey

**Jun Bai**[1]     **Xiaofeng Zhang**[1]     **Chen Li**[1]     **Hanhua Hong**[1]     **Xi Xu**[2]
**Chenghua Lin**[3]     **Wenge Rong**[1]

[1] School of Computer Science and Engineering, Beihang University, China
[2] Faculty of Information Technology, Beijing University of Technology, China
[3] Department of Computer Science, University of Manchester, United Kingdom
{ba1_jun, xiaofeng_z, chen.li, hanhua_hong, w.rong}@buaa.edu.cn, xuxi@bjut.edu.cn
chenghua.lin@manchester.ac.uk

## Abstract

Transferability estimation has been attached to great attention in the computer vision fields. Researchers try to estimate with low computational cost the performance of a model when transferred from a source task to a given target task. Considering the effectiveness of such estimations, the communities of natural language processing also began to study similar problems for the selection of pre-trained language models. However, there is a lack of a comprehensive comparison between these estimation methods yet. Also, the differences between vision and language scenarios make it doubtful whether previous conclusions can be established across fields. In this paper, we first conduct a thorough survey of existing transferability estimation methods being able to find the most suitable model, then we conduct a detailed empirical study for the surveyed methods based on the GLUE benchmark. From qualitative and quantitative analyses, we demonstrate the strengths and weaknesses of existing methods and show that H-Score generally performs well with superiorities in effectiveness and efficiency. We also outline the difficulties of consideration of training details, applicability to text generation, and consistency to certain metrics which shed light on future directions.

## 1   Introduction

Recent advances in the community of Natural Language Processing (NLP) are heavily built on the effectiveness of Pre-trained Language Models (PLMs), especially on large ones (LLMs) (Zeng et al., 2023; OpenAI, 2022; Touvron et al., 2023; Wang et al., 2023). As the number of available PLMs continually grows, a critical question arises: "*Which PLM can make the performance of a downstream task best?*". The fine-tuning result on a task usually varies across different PLMs, and this variation becomes more pronounced in low-resource scenarios (Bassignana et al., 2022).

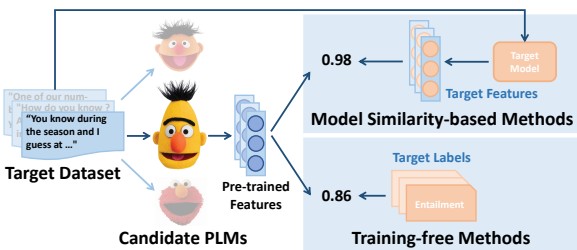

Figure 1: Diagram of transferability estimation methods. Based on the pre-trained features of target samples output from candidate PLM, model similarity-based methods and training-free methods estimate the transferability by inter-model similarity and the compatibility between pre-trained features and target labels.

Basically, the key to such model selection is to figure out the transferability between the model and the target task. Pioneering works conducted fine-tuning on every candidate model in a brute-force manner (Phang et al., 2018; Zamir et al., 2019). Though the true fine-tuning performance can be obtained in this way, expensive parameter optimization is practically prohibitive (Wolf et al., 2020). Thus, there is an urgent need to quantify the transferability at a low cost of computation. To this end, Transferability Estimation (TE), as an essential task of Transfer Learning (TL), has emerged as a key challenge with several solutions proposed in Computer Vision (CV) fields initially (Agostinelli et al., 2022). Recently, some of these remarkable approaches have also been applied to NLP tasks which show promising results on PLM selection (Bassignana et al., 2022; Vu et al., 2022).

Despite a great number of surveys established for TL and PLMs (Niu et al., 2020; Plested and Gedeon, 2022; Guo et al., 2022), there is no comprehensive survey on TE yet, especially with the purpose of PLM selection. Therefore, this paper aims to fill this gap by providing a comprehensive and well-structured summary of recent progress. To ensure comprehensive coverage, a multi-stage approach is employed to identify and select the

studies included in this review. Firstly, an extensive literature search was carried out using online databases, such as Google Scholar and DBLP. The search terms used were carefully chosen to capture the key concepts and themes related to TE and PLMs. After retrieving an initial pool of nearly 100 articles, a thorough screening of titles, abstracts, and keywords was conducted to exclude irrelevant studies, leading to a final selection of 20 studies that met the predetermined criteria for inclusion.

Based on these research, we present a method taxonomy. As shown in Fig. 1, according to the need for training on target task, we divide TE methods into: (1) *Model Similarity-based Methods* that assume the inter-model similarity reflects the transferability which require the model trained on target task (Dwivedi and Roig, 2019). (2) *Training-free Methods* that accelerate the estimation process by computing metrics free of target model training to examine the compatibility of the PLM's feature space on the target dataset (Ding et al., 2022). Then we conduct qualitative analysis for the applicability and provide empirical results on the GLUE benchmark (Wang et al., 2019) to manifest specific strengths and weaknesses in existing methods. We show that model similarity-based methods have the superiority of applicability to different target tasks, and training-free methods have the advantage over fast estimation. And for the methods simulating the dynamics of fine-tuning, they generally perform better. Besides, we analyze some factors that can affect the estimation effectiveness and efficiency including task type, sample size, feature dimension, target model as well as sample affinity function. The empirical observations demonstrate that H-Score (Bao et al., 2019) generally shows desired usability. Based on these investigations, we further exhibit some under-explored aspects to shed light on the future directions [1].

## 2   Related Work

**Transfer Learning.**   Training robust supervised models from scratch is non-trivial, especially in low-resource scenarios (Jin et al., 2023). Aiming at transferring knowledge from a source task to a target task, TL can achieve superior performances on the target dataset by spending far less time and using far fewer data (Niu et al., 2020). Despite a good number of surveys available for TL (Ruder

et al., 2019; Niu et al., 2020; Alyafeai et al., 2020; Iman et al., 2022), these works mainly focus on "*what to transfer?*" and "*how to transfer?*" that describe specific transfer approaches. To the best of our knowledge, there is no comprehensive survey on TE yet, which seeks to answer the question of "*when to transfer?*". This work is expected to fill this gap by shedding light on how to appropriately choose TE methods for PLMs practitioners.

**Transferability Estimation.**   To avoid exhaustive attempts on all pairs of source tasks and target tasks, TE provides efficient heuristics to exhibit the best-performing source task at a minor cost (Agostinelli et al., 2022). Originated in the field of CV, a great number of TE approaches, including model-similarity-based methods (Dwivedi and Roig, 2019), label-comparison-based methods (Tran et al., 2019) and source features-based methods (Ding et al., 2022), etc., have been proposed in the past few years. To adapt such techniques to PLM selection for NLP tasks, Bassignana et al. (2022) found the predictions of LogME can positively correlate with the true performances of candidate PLMs, and Vu et al. (2022) exhibited the model similarity computed by soft prompts reflects the transfer performance across different models. Built on these remarkable researches, we further review the TE methods and provide a comprehensive empirical study of them for PLM selection.

**Pre-trained Language Models.**   From BERT (Devlin et al., 2019) to LLaMA (Touvron et al., 2023), significant efforts have been put into scaling PLMs into LLMs and some abilities such as performing arithmetic, answering questions are emerging simultaneously (Schaeffer et al., 2023). Nevertheless, training and fine-tuning LLMs or even small ones require substantial computational resources which can limit accessibility to these models for researchers and developers with limited resources even with the help of parameter-efficient tuning (Hu et al., 2022). Based on these considerations, the efficient utilization of PLMs is still a problem worth studying. Thus we focus on the selection of PLMs in this work which aims at releasing the computing resources needed for exhaustive fine-tuning.

## 3   Transferability Estimation Taxonomy

### 3.1   Problem Formalization

Formally, given a pool of $L$ candidate PLMs $\{\phi_i\}_{i=1}^{L}$ and a target dataset $\mathcal{D} = \{(x_i, y_i)|x_i \in \mathcal{X}, y_i \in \mathcal{Y}\}_{i=1}^{N}$ containing $N$ samples where each

---

[1]The code is available at https://github.com/ba1jun/model-selection-nlp.

$\phi_i$ can encode the sample to pre-trained feature $\phi_i(x_i)$ (usually the [CLS] embedding), the true performance $T_i(\mathcal{D})$ can be measured by certain evaluation metrics after fine-tuning $\phi_i$ on $\mathcal{D}$ with careful tuning of hyper-parameters. The TE approach should produce a score $S_i(\mathcal{D})$ for each $\phi_i$ to approximate the true fine-tuning performance $T_i(\mathcal{D})$. Intuitively, a well-designed method should return $\{S_i(\mathcal{D})\}_{i=1}^{L}$ that correlates well with $\{T_i(\mathcal{D})\}_{i=1}^{L}$ under an acceptable burden, such that the top-performing PLM can be determined rapidly.

## 3.2 Model Similarity-based Methods

To avoid brute force fine-tuning, the model similarity-based methods are designed based on the assumption that a high similarity between two models correlates with a high degree of transferability between the tasks bonded to the models. To this end, one model $\psi$ fine-tuned on the target task, i.e., the target model, is required to compute its similarity to each candidate PLM. Therefore, the time consumption of fine-tuning can be significantly reduced to $1/L$ of brute force approach extra with a minor cost of model similarity computation.

Currently, the sample features output from models are mainly used to measure the inter-model similarity. Therefore, the target is to design a similarity function to maximize the correlation between fine-tuning performances and similarities between the pre-trained features $\{\phi(x_i)\}_{i=1}^{N}$ and target features $\{\psi(x_i)\}_{i=1}^{N}$. In terms of the similarity computation mechanism, existing functions can fall into *sample-wise similarity functions* and *graph-wise similarity functions*:

**Sample-wise Similarity Functions**  The main idea is to directly compute the similarity between features across models. Under the Direct Similarity Estimation (DSE) (Luo et al., 2022) framework, Vu et al. (2020) compute the affinity between the mean features as the model similarity, i.e., $\mathcal{A}(\sum_i \phi(x_i)/N, \sum_i \psi(x_i)/N)$ where $\mathcal{A}$ is the sample affinity function such as Euclidean and cosine distances, while Luo et al. (2022) utilize averaged sample affinities $\sum_i \mathcal{A}(\phi(x_i), \psi(x_i))/N$.

**Graph-wise Similarity Functions**  In the form of Duality Diagram Similarity (DDS) (Dwivedi et al., 2020) framework, the graph-wise functions first measure the affinities for every sample pair in each model feature space separately, then compute the similarity between the resulting affinity graphs $\mathcal{G}_\phi = (\mathcal{V}_\phi, \mathcal{E}_\phi)$ and $\mathcal{G}_\psi = (\mathcal{V}_\psi, \mathcal{E}_\psi)$ with the

sample features as vertices, e.g., $\mathcal{V}_\phi = \{\phi_i\}_{i=1}^{N}$, and the inter-sample affinities as edges, e.g., $\mathcal{E}_\phi = \{\mathcal{A}(\phi_i, \phi_j)|\phi_i, \phi_j \in \mathcal{V}_\phi\}$. For instance, Representation Similarity Analysis (RSA) (Dwivedi and Roig, 2019), Graph-Based Similarity (GBS) (Chen et al., 2021), Kernel Alignment (KA) (Huang et al., 2021) and Centered Kernel Alignment (CKA) (Kornblith et al., 2019) which are only slightly different in the ways to compute inter-sample affinities and inter-graph similarities.

## 3.3 Training-free Methods

Although model similarity-based methods only need to fine-tune on the target task once, they still require a large load of computational cost. Therefore, the training-free methods try to directly compare the pre-trained features $\{\phi(x_i)\}_{i=1}^{N}$ with the true target labels $\{y_i\}_{i=1}^{N}$ by cheap metrics to further save the estimation time. According to whether directly measure the fine-tuning loss, the metrics can be divided into *class separability metrics* and *loss approximation metrics*.

**Class Separability Metrics**  These metrics intuitively examine whether pre-trained features are easy to separate according to their target labels, and assume that well-separated pre-trained features results in desired fine-tuning performance. Some of these metrics directly measure the separability of static pre-trained features. For example, MSC (Meiseles and Rokach, 2020) uses the mean intra-cluster distance and the mean nearest-cluster distance to quantify the clustering quality of pre-trained features over target classes. Similarly, Puigcerver et al. (2021) rank the candidate PLMs by the test accuracy of $k$NN on pre-trained features via the leave-one-out cross-validation. PARC (Bolya et al., 2021) first computes the pairwise affinities between the pre-trained features of each pair of target samples, which is then compared with the pairwise label affinities of each pair of target samples to quantify the source feature space's fitness on target dataset. And GBC (Pándy et al., 2022) uses the Bhattacharyya coefficient to measure the inter-class overlap of pre-trained features, where higher overlap means poorer separability. To further consider the fine-tuning dynamics by assuming the pre-trained features can be adjusted by an extra linear transformation, Kumari et al. (2022) train a cheap Logistic Regression (LR) model on pre-trained features to estimate how fitting the linearly transformed pre-trained features are for their

| Methods | Input | Task Agnostic | Dynamic Consideration | Free of Training |
|---------|-------|---------------|------------------------|------------------|
| *Model Similarity-based Methods* | | | | |
| DSE (Vu et al., 2020) | $\phi(x), \psi(x)$ | ✓ | ✗ | ✗ |
| DDS (Dwivedi et al., 2020) | $\phi(x), \psi(x)$ | ✓ | ✗ | ✗ |
| *Training-free Methods* | | | | |
| MSC (Meiseles and Rokach, 2020) | $\phi(x), y$ | ✗ | ✗ | ✓ |
| $k$NN (Puigcerver et al., 2021) | $\phi(x), y$ | ✗ | ✗ | ✓ |
| PARC (Bolya et al., 2021) | $\phi(x), y$ | ✗ | ✗ | ✓ |
| GBC (Pándy et al., 2022) | $\phi(x), y$ | ✗ | ✗ | ✓ |
| Logistic (Kumari et al., 2022) | $\phi(x), y$ | ✗ | ✓ | ✓ |
| H-score (Bao et al., 2019) | $\phi(x), y$ | ✗ | ✗ | ✓ |
| Reg. H-score (Ibrahim et al., 2022) | $\phi(x), y$ | ✗ | ✗ | ✓ |
| $\mathcal{N}$LEEP (Li et al., 2021) | $\phi(x), y$ | ✗ | ✗ | ✓ |
| TransRate (Huang et al., 2022) | $\phi(x), y$ | ✗ | ✗ | ✓ |
| LogME (You et al., 2021) | $\phi(x), y$ | ✗ | ✓ | ✓ |
| SFDA (Shao et al., 2022) | $\phi(x), y$ | ✗ | ✓ | ✓ |
| PACTran (Ding et al., 2022) | $\phi(x), y$ | ✗ | ✓ | ✓ |

Table 1: The comparison of the surveyed approaches, where $\phi(x)$, $\psi(x)$ and $y$ denote the feature from candidate PLM, feature from target model and target label; ✓ and ✗ represent whether the method fulfill the property or not.

target classes by LR's test accuracy.

**Loss Approximation Metrics** Based on solid theoretical proof, these metrics try to directly approximate the fine-tuning loss that correlates with the fine-tuning performance well. Inspired by Euclidean information geometry, H-Score (Bao et al., 2019) approximates the optimal log-loss by inter-class variance and feature redundancy that characterize the asymptotic error probability of using pre-trained features to estimate target labels. Ibrahim et al. (2022) then propose regularized H-score which further shrinks the error that occurred when inverting the high-dimensional features using a pseudo-inverse. $\mathcal{N}$LEEP (Li et al., 2021) first uses a Gaussian mixture model to attach a posterior distribution on Gaussian components to each pre-trained feature, then computes the likelihood from posterior distribution to target label to approximate that from pre-trained feature to target label. TransRate (Huang et al., 2022) approximates the correlation between pre-trained features and target labels by Mutual Information (MI) which has been proven an upper bound and a lower bound to the log-likelihood. There are also some metrics that involve fine-tuning dynamics. SFDA (Shao et al., 2022) simulates the dynamics by projecting the pre-trained features using Fisher Discriminant Analysis (FDA) to increase the class separability. Then, it approximates the log-likelihood by Bayes

classification over projected features and also adds a self-challenging module to further measure the ability of the pre-trained models on hard samples. To avoid over-fitting problem of maximum likelihood estimation, LogME (You et al., 2021) turns to approximate the marginalized likelihood of label given pre-trained features over all possible linear transformation. More recently, motivated by learning theory, PACTran (Ding et al., 2022) minimizes the PAC-Bayesian upper bound over the log-loss.

## 4 Qualitative Analysis

To examine the applicability of each method, we qualitatively compare them as shown in Table 1 from the following perspectives: (1) *Task Agnostic*: the method does not require certain target task type; (2) *Dynamic Consideration*: the fine-tuning dynamics of pre-trained features are considered; (3) *Free of Training*: the method does not need fine-tuning on target task.

**Task Agnostic** A widely applicable method should be able to deal with multiple task types such as classification, regression, and generation. Currently, only model similarity-based methods satisfy this property. However, the inter-model similarity is only aware of sample features and does not consider the output space of the task.

| Dataset | \|Train\| | \|Dev\| | Task | Metric |
|---|---|---|---|---|
| *Sentence Classification Tasks* | | | | |
| CoLA | 8.5k | 1k | Acceptability | Mcc. |
| SST-2 | 67k | 872 | Sentiment | Acc. |
| *Paraphrase Tasks* | | | | |
| MRPC | 3.7k | 408 | Paraphrase | Acc. |
| QQP | 364k | 40.4k | Paraphrase | Acc. |
| *Inference Tasks* | | | | |
| MNLI | 393k | 9.8k | NLI | Acc. |
| QNLI | 105k | 5.5k | QA/NLI | Acc. |
| RTE | 2.5k | 277 | NLI | Acc. |
| WNLI | 635 | 71 | Coref./NLI | Acc. |

Table 2: Statistic, task type and metric of GLUE tasks.

**Dynamic Consideration**   Since fine-tuning appropriately adjust the representation of pre-trained features to adapt to the target task, the training dynamics are also a key factor. To date, only Logistic, LogME, SFDA, and PACTran assume that the pre-trained features can be adjusted by a linear transformation, while the fine-tuning process can be more diverse, e.g., adapter tuning (Houlsby et al., 2019) and prompt tuning (Lester et al., 2021), how to simulate different dynamics is under-explored.

**Free of Training**   The trained target model is mainly needed by model similarity-based methods. Although this is not a serious problem since the target model can be reused when estimating for different PLMs, it still takes a training time and whether model similarity-based methods perform stably on different target models is also doubtful.

## 5   Experimental Setup

**Datasets**   Following You et al. (2021), we validate TE methods on the GLUE benchmark (Wang et al., 2019) that is a collection of diverse Natural Language Understanding (NLU) tasks. Note, we remove the STS-B task since it is a regression task that is not suitable to most TE methods, the details of others are reported in Table 2.

**Candidate PLMs**   Since the HuggingFace Model Hub (Wolf et al., 2020) has already provided carefully tuned results on GLUE tasks for some models, we select 6 PLMs that have all the GLUE tasks' results following You et al. (2021): namely *bert-base-uncased*, *bert-base-cased* (Devlin et al., 2019), *roberta-base* (Liu et al., 2019) and their distilled versions which are *distilbert-base-uncased*, *distilbert-base-cased* and *distilroberta-base* (Sanh

et al., 2019) whose fine-tuning performances are as shown in Appendix A.

**Methods Setups**   Each candidate PLM to be fine-tuned is followed by a classification layer which takes the output embedding of BOS (Beginning of Sequence, e.g., [CLS] for BERT and  for RoBERTa) token as input, and all the TE methods utilize the same BOS embeddings as pre-trained features. Before estimating by certain methods, we also conduct dimensionality reduction on the sample features by Principal Component Analysis (PCA) (Roweis, 1997) since some methods' performances heavily depend on the number of feature dimensions. For model similarity-based methods, we also try different PLMs to train the target model (i.e., ALBERT: *albert-base-v2*, DeBERTa: *deberta-base* and ELECTRA: *electra-base-discriminator*). Moreover, since some methods need to compute affinity graph which can be very time-consuming when there are too many samples, i.e., DDS, $k$NN, MSC, PARC, LFC, we limit the maximum number of samples that can be used by them to 10k. We run each method 5 times with different random seeds and report the mean results. For the implementation details, please refer to Appendix B.

**Evaluation**   To measure the deviation of TE methods' predictions to true fine-tuning performances, we use MRR and mean Spearman's rank correlation ($\mu_\rho$) on all GLUE tasks. Among them, MRR reveals the average ranking of the best-performing PLM, and $\mu_\rho$ evaluates the overall correlation between predicted score list $\{S_i(\mathcal{D})\}_{i=1}^L$ and true performance list $\{T_i(\mathcal{D})\}_{i=1}^L$. Besides, the time consumption is measured by mean training time ($\mu_{tt}$) that records the time of target model training and mean estimating time ($\mu_{et}$) that tells the wall clock time of estimation value computing, which are both averaged over all GLUE tasks. Note that we omit the time of sample features encoding since this is the same across all methods.

## 6   Quantitative Analysis

**Effectiveness and Efficiency**   The overall metric scores of all methods are reported in Table 3. For estimation effectiveness, although model similarity-based methods excel at adapting to different target tasks, they generally perform worse than training-free methods. Notably, DSE and PARC achieve the best MRR and $\mu_\rho$ respectively, while they need to carefully select sample affinity function to produce

| Method | MRR(↑) | | | | $\mu_\rho$(↑) | | | | $\mu_{tt}$(↓) | $\mu_{et}$(↓) |
|---|---|---|---|---|---|---|---|---|---|---|
| | Sen. | Para. | Infer. | Overall | Sen. | Para. | Infer. | Overall | | |
| *Model Similarity-based Methods* | | | | | | | | | | |
| DSE$_{\text{ALBERT}}$ | 1.00 | 1.00 | 0.88 | 0.94 | 0.49 | 0.54 | 0.40 | 0.45 | 2.4h | 10.1s |
| DSE$_{\text{DeBERTa}}$ | 1.00 | 1.00 | 1.00 | 1.00 | 0.46 | 0.43 | 0.48 | 0.46 | 1.8h | 7.2s |
| DSE$_{\text{ELECTRA}}$ | 1.00 | 1.00 | 1.00 | 1.00 | 0.40 | 0.43 | 0.49 | 0.45 | 1.2h | 9.5s |
| DDS$_{\text{ALBERT}}$ | 0.20 | 0.50 | 0.58 | 0.47 | -0.34 | 0.09 | 0.42 | 0.14 | 2.4h | 131.7s |
| DDS$_{\text{DeBERTa}}$ | 0.42 | 0.50 | 0.56 | 0.51 | 0.20 | 0.40 | 0.31 | 0.32 | 1.8h | 132.8s |
| DDS$_{\text{ELECTRA}}$ | 0.42 | 0.75 | 0.83 | 0.71 | 0.20 | 0.20 | 0.29 | 0.25 | 1.2h | 133.5s |
| *Training-free Methods* | | | | | | | | | | |
| MSC | 0.58 | 0.33 | 0.55 | 0.50 | 0.03 | 0.03 | 0.42 | 0.22 | - | 4.5s |
| $k$NN | 0.33 | 0.67 | 0.79 | 0.65 | 0.26 | 0.40 | 0.49 | 0.41 | - | 21.0s |
| PARC | 0.60 | 1.00 | 1.00 | 0.90 | 0.69 | 0.69 | 0.66 | 0.67 | - | 87.5s |
| GBC | 0.33 | 0.67 | 0.38 | 0.44 | -0.14 | 0.54 | 0.24 | 0.22 | - | 0.6s |
| Logistic | 0.25 | 1.00 | 1.00 | 0.81 | 0.31 | 0.46 | 0.67 | 0.53 | - | 5.1s |
| H-Score | 0.60 | 1.00 | 0.79 | 0.80 | 0.46 | 0.57 | 0.64 | 0.57 | - | 37.5s |
| Reg. H-Score | 0.60 | 1.00 | 1.00 | 0.90 | 0.49 | 0.71 | 0.61 | 0.60 | - | 33.4s |
| $\mathcal{N}$LEEP | 0.33 | 0.67 | 1.00 | 0.75 | -0.11 | 0.46 | 0.61 | 0.39 | - | 17.0s |
| TransRate | 0.17 | 0.75 | 0.25 | 0.35 | -0.69 | 0.57 | 0.12 | 0.03 | - | 0.7s |
| LogME | 0.58 | 1.00 | 1.00 | 0.90 | 0.34 | 0.66 | 0.74 | 0.62 | - | 10.9s |
| SFDA | 0.50 | 0.75 | 0.81 | 0.72 | 0.71 | 0.54 | 0.54 | 0.59 | - | 116.5s |
| PACTran | 0.58 | 1.00 | 0.75 | 0.77 | 0.29 | 0.63 | 0.56 | 0.51 | - | 7.2s |

Table 3: The performance of TE methods on GLUE benchmark, where MRR, mean Spearman coefficient $\mu_\rho$ on each type of task (Sen. for single sentence classification, Para. for paraphrase, and Infer. for inference) and all tasks (Overall) are reported. Moreover, mean training time $\mu_{tt}$ and mean estimating time $\mu_{et}$ are listed to show the method efficiency where "-" means not applicable. For model similarity-based methods, the subscript indicates the type of its target model, e.g., DSE$_{\text{ALBERT}}$ means DSE implemented with ALBERT as target model.

such desired results. Another obvious observation is that Logistic, LogME, SFDA, and PACTran all result in a remarkable performance, which empirically validates the importance of fine-tuning dynamics. For estimation efficiency, model similarity-based methods all need considerable time consumption on target model training. Unless the target model training time is negligible compared to the whole PLM selection time, this kind of method has no advantage over the training-free method in terms of speed. Generally, the training-free methods run pretty fast, only the methods need to compute affinity graph will consume a lot of time ($k$NN, MSC, LFC, PARC), which have to limit the amount of data samples when they are employed on huge datasets. For detailed results on each task, please refer to Appendix C.

**Sensitivity to Task Type** More detailed performances on three different types of tasks are also reported in Table 3. Generally, TE methods perform better on sentence pair tasks (paraphrase and inference) than on single sentence tasks (sentence classification), where 11 out of 14 TE methods re-

sult in $\mu_\rho$ of sentence pair tasks superior to that of single sentence tasks. We speculate the reason is that most candidate PLMs used in this work are from the BERT family and the corresponding Next Sentence Prediction (NSP) (Devlin et al., 2019) pre-training task makes the [CLS] embedding more suitable for encoding a sentence pair, such that TE methods cannot fully understand the samples through the pre-trained features when meeting sentence classification task.

**Robustness to Fewer Samples** The key strength of training-free methods lies in no need to target model training, while the total time consumption can be further reduced if the method also performs well when using only a small amount of data samples, such that the encoding time of ignored samples can be saved (Encoding for all GLUE datasets takes 1.73 hours in our case). To examine the data efficiency, we select 8 top-performing training-free methods and conduct PLM selection when different percentages of data are conditioned. Figure 2 illustrates the overall performance variation on GLUE

| Method | Euclidean | | | Cosine | | | Correlation | | |
|---|---|---|---|---|---|---|---|---|---|
| | MRR($\uparrow$) | $\mu_\rho(\uparrow)$ | $\mu_{et}(\uparrow)$ | MRR($\uparrow$) | $\mu_\rho(\uparrow)$ | $\mu_{et}(\downarrow)$ | MRR($\uparrow$) | $\mu_\rho(\uparrow)$ | $\mu_{et}(\downarrow)$ |
| DSE | 1.00 | 0.46 | 7.2s | 0.46 | 0.26 | 7.3s | 0.44 | 0.20 | 9.2s |
| DDS | 0.50 | -0.10 | 129.4s | 0.51 | 0.31 | 133.1s | 0.51 | 0.32 | 135.9s |
| $k$NN | 0.50 | 0.22 | 1.7s | 0.59 | 0.34 | 8.4s | 0.65 | 0.41 | 21.0s |
| LFC | 0.41 | -0.01 | 6.7s | 0.45 | 0.09 | 8.6s | 0.39 | 0.08 | 8.7s |
| PARC | 0.69 | 0.35 | 83.6s | 0.90 | 0.64 | 83.8s | 0.90 | 0.67 | 87.5s |
| MSC | 0.61 | 0.08 | 4.5s | 0.50 | 0.22 | 4.6s | 0.50 | 0.22 | 37.8s |

Table 4: The performance comparison of methods on GLUE benchmark when different affinity functions (Euclidean distance, cosine distance, and correlation distance) are employed, where the MRR score, mean Spearman coefficient $\mu_\rho$ and mean estimating time $\mu_{et}$ are reported.

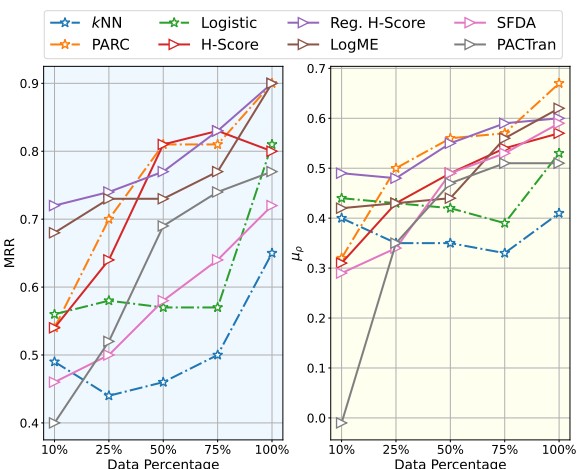

Figure 2: The performance variation on GLUE tasks of training-free methods when different percentages of target data samples are used to conduct the transferability estimation, in which the original class proportion will be kept when the sub-dataset is sampled.

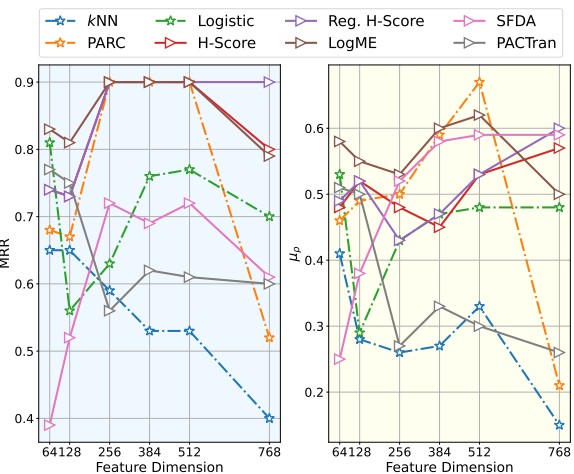

Figure 3: The performance variation on GLUE tasks of training-free methods when pre-trained features' dimensions are reduced to different lengths by PCA.

of training-free methods as the data percentage changed. By employing a shrinkage-based estimator of covariance, regularized H-Score exhibits stable performance compared to the original H-score. And LogME also shows similar stability on both MRR and $\mu_\rho$ since it is based on the marginalized likelihood which can alleviate the over-fitting problem on a small dataset. We can observe that $k$NN also performs stable on $\mu_\rho$ while it needs careful selections of $k$ and sample affinity function. However, a significant decrease in MRR occurred on almost all methods, indicating more advanced approaches are required to achieve accurate selection in low-resource scenarios.

**Effect of Feature Dimensions** If a method performs best when conducted on the pre-trained features with the original dimension, then there is no need to employ dimensionality reduction and tune the reduced dimensions, which can further save the

estimating time. As shown in Figure 3, we list the performance variation of top-8 performing training-free methods on different feature dimensions. It is observed that H-Score and regularized H-Score preferably enjoy original dimensions because the original feature space helps them to approximate the feature redundancy better, while the others all achieve the best results on smaller dimensions. For $k$NN, PARC, they need to measure sample affinity which may encounter the curse of dimensionality in high-dimensional scenes, thus performing better when the feature dimension is small meanwhile dimensionality reduction will not lose too much original information. For Logistic, LogME, SFDA, and PACTran which assume the pre-trained features can be linearly transformed, eliminating redundant feature dimensions can results in better estimation results.

**Sensitivity to Different Target Models** Although model similarity-based methods only need

to train one target model, we actually have rich choices of PLM to train target models. An ideal method should produce similar results when different target models are implemented such that we can save the time required to try different target models. To examine the sensitivity to the target model, we conduct model similarity-based methods with different target models and show the results in Table 3. We can observe different behaviors in which DSE performs stably while DDS is very sensitive to the type of target model. Since the main difference between DSE and DDS is that the former computes inter-sample affinities across models while the latter compares the affinity graph across models, this observation reflects that the affinity graph from the target model may not well reflect the target task mechanism and directly measuring the affinity between features across models is preferable.

**Effect of Sample Affinity Function** As introduced in Section 3, the implementation of DSE, DDS, $k$NN, MSC, LFC, and PARC require certain sample affinity functions. We try Euclidean, cosine, and correlation distances for the above methods to examine whether different functions will affect the methods and report the results in Table 4. Compared to Euclidean distance, cosine distance and correlation distance conduct extra normalization operations and thus results in more estimating time. However, except when applied to DSE, cosine and correlation distances generally exhibit superior performance than Euclidean distance. This observation reveals that the norm of the feature vector may result in anisotropic feature space and should not be taken into account to measure the sample affinity, which is also supported by (Su et al., 2021) that suggests the normalization operation can alleviate the anisotropy problem of PLMs.

## 7 Conclusion and Future Directions

This paper reviews the recent advances in TE that can be applied to PLM selection and presents a method taxonomy based on a thorough analysis. Moreover, comprehensive qualitative and quantitative comparisons between different approaches are provided to help understand their applicability in a number of aspects. We hope this survey can help people for the purpose of research or industry to choose desired PLM by appropriate TE methods.

Although lots of efforts have been made as surveyed, there still remain some directions that deserve further investigation:

(1) **How to make the estimation approach aware of fine-tuning strategies and experimental hyper-parameters?** The fine-tuning strategy usually needs to be determined under the acceptable computation burden, i.e. fully fine-tuning (optimizing all model parameters) or parameter-efficient tuning (optimizing part of model parameters). The actual fine-tuning strategy not only affects the training time and computation consumption but also the loss landscape of PLM which results in different target task performance (Bassignana et al., 2022). However, current approaches usually consider the situation of one strategy whose effectiveness can not be guaranteed in other situations. Therefore, making the estimation able to adapt to different fine-tuning strategies is worth further exploring. Besides, even if the best PLM can be accurately selected, one still needs exhaustive searching of training hyper-parameters to produce desired fine-tuning performance. It is also interesting to consider other important hyper-parameters such as learning rate and temperature when estimating the transferability.

(2) **How to adapt TE methods to text generation task?** Although model similarity-based methods do not assume the type of target task since these methods only rely on the sample features, they neglect the information of the target label and thus the mapping from input space to output space is not well captured and the corresponding task can not be fully understood. However, taking label information into consideration for the text generation task is challenging since the length of output text changes and the one-to-many issue exists (Bao et al., 2020; Zheng et al., 2021; Zhao et al., 2023). Since currently LLMs conduct all tasks in the way of text generation and the number of LLMs is continually increasing, the TE method tailored for text generation is urgently needed.

(3) **How to make estimation results consistent with specific evaluation metrics?** In our experiments, the TE methods are asked to correlate just one evaluation metric for GLUE datasets, e.g., Acc for QNLI. However, some tasks may have diverse metrics, e.g., NDCG, R@1 for ranking tasks, and sometimes one may focus on one of the metrics and the variations of these metrics are not necessarily consistent, such that the TE method's predictions can be confusing in these cases. Therefore, how to make TE methods aware of our interested metric is another direction worth exploring.

## Limitations

This work provides a comprehensive summary of existing TE methods. However, limited by our experimental conditions, we have to examine surveyed methods on a toy experimental setting where the following problems need to be improved: (1) Only small-scale PLMs form the candidate pool, the effectiveness of TE methods to select the best-performing LLM is needed to be verified given the current popularity of LLMs. (2) Since most of existing TE methods only support target task of classification type, we determine the GLUE benchmark as the evaluation datasets, while the TE performances on regression task, structure prediction task and generation task are still under-explored.

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

## A Fine-tuning Results on GLUE

| PLM Candidates | CoLA | SST-2 | MRPC | QQP | MNLI | QNLI | RTE | WNLI |
|---|---|---|---|---|---|---|---|---|
| bert-base-uncased | 56.3 | 92.7 | 88.6 | 89.6 | 84.7 | 91.8 | 69.3 | 53.5 |
| distilbert-base-uncased | 51.3 | 91.3 | 87.5 | 88.5 | 82.2 | 89.2 | 59.9 | 56.3 |
| bert-base-cased | 58.2 | 91.7 | 87.8 | 89.2 | 83.9 | 91.0 | 66.1 | 46.5 |
| distilbert-base-cased | 47.2 | 90.4 | 85.6 | 87.8 | 81.5 | 88.2 | 60.6 | 56.3 |
| roberta-base | 63.6 | 94.8 | 90.2 | 91.9 | 87.6 | 92.8 | 78.7 | 57.7 |
| distilroberta-base | 59.3 | 92.5 | 86.6 | 89.4 | 84.0 | 90.8 | 67.9 | 52.1 |

Table 5: The best fine-tuning performances of candidate PLMs on GLUE dev datasets reported from HuggingFace, where the metrics are Matthews Correlation Coefficient (MCC) for CoLA and Accuracy for the other datasets.

## B Implementation Details

Our experimental machine contains an Intel(R) Xeon(R) Silver 4110 CPU @ 2.10GHz and a NVIDIA GeForce RTX 3090 24G GPU. For the implementation of TE methods, since some methods' performance heavily depend on the number of feature dimensions, we reduce the feature dimension to [16, 32, 64, 128, 256, 512, 768] by PCA for each method to find their most suitable dimensions. For DSE, RSA, kNN, MSC, LFC, PARC that need to compute the affinities between sample features, we also try different sample affinity functions including cosine, euclidean and correlation distances. The implementation details of surveyed methods are as follows:

**DSE** Among the averaged sample affinities (Luo et al., 2022) and the affinity between the mean features (Vu et al., 2020), we found that the former performs better. And DSE achieves the best results when DeBERTa is taken as target model, Euclidean distance is used to measure the sample affinity and the number of feature dimensions is 768. The corresponding computation is as Eq. 1.

$$\mathcal{S}(\mathcal{D}) = -\frac{1}{N} \sum_{i=1}^{N} \|(\phi(x_i) - \psi(x_i)\|_2 \qquad (1)$$

**DDS** Among a number of instances of DDS framework (Dwivedi et al., 2020), RSA (Dwivedi and Roig, 2019) shows the best performance when DeBERTa is trained as target model, correlation distance is used and the number of feature dimensions is 512. Specifically, the pre-trained features and target features are first processed by z-score normalization. Then the pre-trained affinity graph and target affinity graph are computed by correlation distance. Finally, the lower triangular adjacent matrices of two graphs are compared by Spearman correlation coefficient as transferability score.

**MSC** We use the code of silhouette_score from scikit-learn to implement MSC, which exhibits the best performance when cosine distance is used and the number of feature dimensions is 256.

**kNN** We use the code of KNeighborsClassifier from scikit-learn and use the test accuracy of leave-one-out cross-validation to quantify the transferability. We tune the $k$ in [1, 3, 5, 7], the method exhibits the best performance when $k = 5$, correlation distance is used and the number of feature dimensions is 64.

**PARC** The computation process of PARC is similar to that of RSA except that the target affinity graph is replaced by affinity graph derived from samples' one-hot labels. We use the code from here and the best performance is achieved when correlation distance is used and the number of feature dimensions is 512.

**GBC** It first uses Gaussian distribution to model each target class of samples which is parameterized by the in-class pre-trained features vectors. Then the averaged Bhattacharyya distance between every pair of different classes are used to measure the inter-class overlap as Eqs. 2 and 3:

$$\mathrm{BC}(p_{v_i}, p_{v_j}) = \int \sqrt{p_{v_i}(\phi(x)) p_{v_j}(\phi(x))} dx \qquad (2)$$

$$\mathcal{S}(\mathcal{D}) = -\sum_{i \neq j} \mathrm{BC}(p_{v_i}, p_{v_j}) \tag{3}$$

where $v$ is a specific value of target classes. We use the code from here and the most suitable number of feature dimensions is 64.

**Logistic**    We use the code of LogisticRegression from scikit-learn with the default hyper-parameters to classify the pre-trained features. The test accuracy of leave-one-out cross-validation is used to quantify the transferability and the most suitable number of feature dimensions is 64.

**H-Score**    As Eq. 4 shows, it first computes the covariance matrix over the feature dimensions of pre-trained features and that over the feature dimensions of each target class's mean feature, then the trace of the dot-product between the inverse matrix of former and the latter is used to approximate the optimal log-loss. We use the code from here and the most suitable number of feature dimensions is 768.

$$\mathcal{S}(\mathcal{D}) = \mathrm{tr}(\mathrm{cov}(\phi(\mathcal{X}))^{-1}\mathrm{cov}(\mathbb{E}_{P(\mathcal{X}|\mathcal{Y})}[\phi(\mathcal{X})|\mathcal{Y}])) \tag{4}$$

**Regularized H-Score**    Compared to H-Score, it further solve the statistical problem of covariance estimation by shrinkage-based estimator (Ibrahim et al., 2022). The most suitable number of feature dimensions is 768.

$\mathcal{N}$**LEEP**    It first uses Gaussian mixture model to fit the pre-trained features, then computes the Log Expected Empirical Prediction (LEEP) score between posterior distribution derived from fitted Gaussian mixture model and the target labels as Eq. 5:

$$\mathcal{S}(\mathcal{D}) = \frac{1}{N}\sum_{i=1}^{N}\log(\sum_{c \in \mathcal{C}} P(y|c)P(c|\phi(x))) \tag{5}$$

where $c$ is the specific Gaussian component and $\mathcal{C}$ is the space of all components. We use the code from here where the number of Gaussian components is five times that of target classes and the most suitable number of feature dimensions is 64.

**TransRate**    It argues that the mutual information $I(\phi(\mathcal{X}), \mathcal{Y}) = H(\phi(\mathcal{X})) - H(\phi(\mathcal{X})|\mathcal{Y})$ between the pre-trained features and the target labels serves as a strong indicator for the performance of model. Since the mutual information is notoriously difficult to compute especially for continuous variables in high-dimensional settings, the authors turn to utilize rate distortion that is closely related to Shannon entropy. The code can be found in (Huang et al., 2022) and the most suitable number of feature dimensions before computing the mutual information is 64.

**LogME**    It uses marginal evidence of the target task $P(y|\phi(x)) = \int P(w)P(y|\phi(x), w)dw$ where $w$ is the weight of linear classifier. The prior $P(w)$ is defined as Gaussian and $P(y|\phi(x), w)$ is a Gaussian likelihood, then $P(y|\phi(x))$ can be analytically estimated. We use the code from here, and the most suitable number of feature dimensions is 512.

**SFDA**    It first projects the pre-trained features by regularized Fisher Discriminant Analysis such that the projected features can have better separability of target classes which can simulate the fine-tuning dynamics. Based on Bayes classification, the projecting weights can be used to compute the prediction probability of each sample on target label which is further used to compute the log-likelihood as the transferability. Moreover, confidential mix noise is further added to examine the model's ability to classify hard samples. We use the code from here and the most suitable number of feature dimensions is 512.

**PACTran**    It has three instances using the Dirichlet, Gamma and Gaussian as prior distributions respectively. Since the first two priors require the pre-training task head layer or non-negative features which are not the case in PLM selection, we implement PACTran with the more general Gaussian prior by the code from here, and tune the $\lambda$ and $\sigma_0^2$ in [0.1, 1, 10] and [1, 10, 100, 1000]. The PACTran performs best when $\lambda = 1, \sigma_0^2 = 10$ and the number of feature dimensions is 64.

## C   TE Methods Performances on GLUE Tasks

| Method | CoLA | SST-2 | MRPC | QQP | MNLI | QNLI | RTE | WNLI |
|---|---|---|---|---|---|---|---|---|
| *Reciprocal Rank* | | | | | | | | |
| DSE | 1.00 | 1.00 | 1.00 | 1.00 | 1.00 | 1.00 | 1.00 | 1.00 |
| DDS | 0.50 | 0.33 | 0.50 | 0.50 | 0.50 | 1.00 | 0.50 | 0.25 |
| MSC | 1.00 | 0.17 | 0.50 | 0.17 | 1.00 | 0.50 | 0.50 | 0.20 |
| *k*NN | 0.33 | 0.33 | 0.33 | 1.00 | 1.00 | 1.00 | 0.17 | 1.00 |
| PARC | 1.00 | 0.20 | 1.00 | 1.00 | 1.00 | 1.00 | 1.00 | 1.00 |
| GBC | 0.50 | 0.17 | 0.33 | 1.00 | 0.50 | 0.33 | 0.50 | 0.20 |
| Logistic | 0.33 | 0.17 | 1.00 | 1.00 | 1.00 | 1.00 | 1.00 | 1.00 |
| H-Score | 1.00 | 0.20 | 1.00 | 1.00 | 1.00 | 1.00 | 1.00 | 0.17 |
| Reg. H-Score | 1.00 | 0.20 | 1.00 | 1.00 | 1.00 | 1.00 | 1.00 | 1.00 |
| NLEEP | 0.50 | 0.17 | 0.33 | 1.00 | 1.00 | 1.00 | 1.00 | 1.00 |
| TransRate | 0.17 | 0.17 | 0.50 | 1.00 | 0.25 | 0.25 | 0.25 | 0.25 |
| LogME | 1.00 | 0.17 | 1.00 | 1.00 | 1.00 | 1.00 | 1.00 | 1.00 |
| SFDA | 0.50 | 0.50 | 0.50 | 1.00 | 1.00 | 1.00 | 1.00 | 0.25 |
| PACTran | 1.00 | 0.17 | 1.00 | 1.00 | 1.00 | 0.50 | 1.00 | 0.50 |
| *Spearman correlation coefficient* | | | | | | | | |
| DSE | 0.43 | 0.49 | 0.37 | 0.49 | 0.49 | 0.31 | 0.37 | 0.75 |
| RSA | 0.49 | -0.09 | 0.20 | 0.60 | 0.26 | 0.71 | 0.43 | -0.06 |
| MSC | 0.94 | -0.89 | 0.31 | -0.26 | 0.60 | 0.71 | 0.77 | -0.41 |
| *k*NN | 0.60 | -0.09 | 0.37 | 0.43 | 0.71 | 0.94 | -0.26 | 0.57 |
| PARC | 0.94 | 0.43 | 0.54 | 0.83 | 0.77 | 0.60 | 0.89 | 0.38 |
| GBC | 0.60 | -0.89 | 0.37 | 0.71 | 0.37 | 0.49 | 0.77 | -0.67 |
| Logistic | 0.49 | 0.14 | 0.14 | 0.77 | 0.71 | 0.71 | 0.60 | 0.64 |
| H-Score | 0.77 | 0.14 | 0.49 | 0.66 | 0.83 | 0.83 | 0.94 | -0.06 |
| Reg. H-Score | 0.83 | 0.14 | 0.77 | 0.66 | 0.83 | 0.83 | 0.71 | 0.06 |
| NLEEP | 0.66 | -0.89 | 0.37 | 0.54 | 0.60 | 0.83 | 0.20 | 0.81 |
| TransRate | -0.89 | -0.49 | 0.77 | 0.37 | 0.09 | 0.43 | -0.31 | 0.26 |
| LogME | 0.83 | -0.14 | 0.66 | 0.66 | 0.77 | 0.83 | 0.83 | 0.52 |
| SFDA | 0.94 | 0.49 | 0.37 | 0.71 | 0.83 | 0.89 | 0.60 | -0.14 |
| PACTran | 0.77 | -0.20 | 0.66 | 0.60 | 0.71 | 0.77 | 0.66 | 0.09 |
| *Estimating Time* | | | | | | | | |
| DSE | 0.38s | 2.90s | 0.16s | 17.86s | 31.86s | 4.48s | 0.09s | 0.03s |
| RSA | 157.62s | 221.01s | 25.11s | 224.61s | 226.94s | 220.50s | 10.58s | 0.63s |
| MSC | 4.94s | 7.30s | 0.98s | 7.32s | 7.22s | 7.35s | 0.48s | 0.06s |
| *k*NN | 2.59s | 2.52s | 0.51s | 117.75s | 28.28s | 15.99s | 0.26s | 0.04s |
| PARC | 104.13s | 145.10s | 17.32s | 141.21s | 142.70s | 142.18s | 7.31s | 0.45s |
| GBC | 0.07s | 0.34s | 0.02s | 1.70s | 1.85s | 0.50s | 0.01s | 0.01s |
| Logistic | 0.55s | 3.28s | 0.31s | 13.35s | 20.02s | 3.19s | 0.19s | 0.09s |
| H-Score | 3.04s | 11.78s | 2.03s | 125.86s | 136.40s | 17.47s | 1.84s | 1.32s |
| Reg. H-Score | 3.84s | 17.15s | 2.55s | 95.83s | 118.07s | 25.83s | 2.30s | 1.74s |
| NLEEP | 18.49s | 27.26s | 4.83s | 26.96s | 32.49s | 22.80s | 3.22s | 0.27s |
| TransRate | 0.09s | 0.42s | 0.02s | 2.16s | 2.33s | 0.69s | 0.02s | 0.01s |
| LogME | 1.23s | 5.07s | 0.69s | 37.80s | 33.36s | 7.55s | 0.62s | 0.55s |
| SFDA | 9.37s | 67.36s | 4.22s | 356.30s | 385.94s | 104.57s | 3.07s | 1.38s |
| PACTran | 0.68s | 4.37s | 0.36s | 19.52s | 27.23s | 4.82s | 0.20s | 0.09s |

Table 6: The reciprocal rank scores, Spearman correlation coefficients and estimating time of TE methods on each GLUE dataset.