# OpenReview forum: "How to Determine the Most Powerful Pre-trained Language Model without Brute Force Fine-tuning? An Empirical Survey"
_EMNLP/2023/Conference — EMNLP 2023 Findings_

### Official Review · Reviewer_35ZS · 2023-07-19

**Soundness:** 3

**Excitement:**

4: Strong: This paper deepens the understanding of some phenomenon or lowers the barriers to an existing research direction.

**Missing References:**

110: For transfer learning, I'd probably include work by Ruder et al. https://aclanthology.org/N19-5004/



**Paper Topic And Main Contributions:**

This paper presents a survey of transferability estimation to answer the question about which pre-trained language model should be used to maximise the performance for a specific downstream task. To that end, the paper compares 14 different methods and evaluates them with the help of six different pre-trained language models. As such, this survey presents a model selection approach which has the potential to contribute help and guide research in identifying suitable models for suitable tasks.

**Questions For The Authors:**

Question A: The models you compared seem to be quite similar. Would you think that including other models, like e.g. T5 could have changed results? What about LLMs?

Question B: Basically, you want to find out which models are similar, right? Because we can expect similar models to perform similarly on the same tasks, correct? This might be a shot in the dark, but have you considered the alignment of different models' feature spaces? I think this could bring in an interesting (and further) dimension of transferability estimation (my hypothesis being that models with similar feature spaces perform similarly).



**Reasons To Accept:**

The following are reasons to accept:

- The paper presents an extensive evaluation of methods for transferability estimation and thus fills a gap in this specific research area.

- The authors present a well-researched and mostly well-written survey. They present the issue and the different parts in digestible sections and follow a clear and reasonable methodology.

- The authors draw valuable conclusions from their work and present interesting future directions.

- The ablation study is particularly interesting, as is the feature dimension reduction.

**Reasons To Reject:**

There are per se no reasons to reject this paper. However, there are a few points which could improve the paper:

- include some examples in the problem formalisation. this is a bit "dry" to read.

- the paper could have benefited from a more diverse model selection. See also Q1.

**Reproducibility:**

4: Could mostly reproduce the results, but there may be some variation because of sample variance or minor variations in their interpretation of the protocol or method.

**Reviewer Confidence:**

3: Pretty sure, but there's a chance I missed something. Although I have a good feel for this area in general, I did not carefully check the paper's details, e.g., the math, experimental design, or novelty.

**Typos Grammar Style And Presentation Improvements:**

I include general questions here. I do not expect a detailed answer for them, but are thought as food for thought for the authors. Some comments are high-level and I leave it to the authors if they want to consider them. Some are just common thoughts.

015: "make a survey" does not sound like an idiomatic expression

019: based on the GLUE

033: especially on Large Language Models (LLMs)

037: clear formulation of the problem --> good!

040: variation goes more significant --> reformulate

042: delete such

044: work

050: impossible rather than prohibitive

050: Thus,

052: what problem was solved --> maybe give an example?

105: the H-Score

139: explain LogME

144: in what way are the researches remarkable?

Section 3: While the problem formulation and formalisation are correct, it would help to have examples instead of mathematical expressions only.

231: the target task

274: the H-Score

Table 1: good overview, maybe colorise hooks (blue) and crosses (red) to make them more distinguishable

327: are also

Table 2: explain Mcc.

Appendix A/Table 5: highlight best-performing score per dataset/task

887: As Eq. 4 shows,

385: MRR?

---

> ### Author Rebuttal · Authors · 2023-08-23
>
> Thanks for you kind comments!
>
> (1) We will polish the problem formalisation to facilitate the understanding of readers.
>
> (2) Answer to question A: We will explore more challenging scenarios of model selection including LLMs which also face the selection problem as the increasing number of LLMs.
>
> (3) Answer to question B: Thanks for you suggestion! The similarity between target model and candidate model has been utilized by Model Similarity-based Methods to score the transferability of candidate model, while training an extra target model might be time-consuming.  However, the utilization of similarity between candidate models is unexplored in Training-free Methods. As you pointed, I totally agree this is an impoortant consideration worthy of further exploration.
>
> (4) We will add the missing references.
>
> (5) Thank you for your patience in pointing out typos and writing improvements which made us learn a lot, we will improve these points in the revised version.

---

### Official Review · Reviewer_oqBj · 2023-08-04

**Typos Grammar Style And Presentation Improvements:** This paper needs to be proofread by a…
**Soundness:** 2

**Excitement:**

2: Mediocre: This paper makes marginal contributions (vs non-contemporaneous work), so I would rather not see it in the conference.

**Paper Topic And Main Contributions:**

The authors conduct a qualitative and quantitative analysis of many transferability estimation methods in the context of pre-trained language models. The paper's text compares and contrasts the methods while tables and graphs present experimental results for running the methods with several publicly available models and the GLUE benchmark. As a result, readers can gain a sense of the relative advantages and disadvantages of different transferability estimation methods for these types of language models and NLP tasks.

**Reasons To Accept:**

The paper's primary results represent a significant amount of work that could be useful to the community. Table 1 provides a compact comparison of many transferability estimation methods with references. Table 3 shares the results of an extensive comparison that highlights advantages and disadvantages of different methods. These two tables could be a good starting point for community members interested in getting started with transferability estimation for language models.

**Reasons To Reject:**

While the paper's experimental results are potentially valuable, there are some major issues with the writing. One challenge of writing a survey paper is clearly and succinctly describing a range of existing work while adding useful cross-work comparisons. The survey should provide more value than a list of titles and abstracts. The current draft of this paper does not meet the standards of academic writing or fluent English. Descriptions and comparisons of methods are not presented clearly in a way that enables linear reading.

Recommendation to the authors: Ask a fluent English speaker who has published papers at past ACL/EMNLP conferences for help revising the paper (including both organization and presentation of information).

**Reproducibility:**

4: Could mostly reproduce the results, but there may be some variation because of sample variance or minor variations in their interpretation of the protocol or method.

**Reviewer Confidence:**

4: Quite sure. I tried to check the important points carefully. It's unlikely, though conceivable, that I missed something that should affect my ratings.

---

> ### Author Rebuttal · Authors · 2023-08-23
>
> Thanks for you comments! We will polish the paper to improve our writing to meet the standard of academic publication.

---

### Official Review · Reviewer_JTER · 2023-08-11

**Typos Grammar Style And Presentation Improvements:** N/A
**Soundness:** 3

**Excitement:**

2: Mediocre: This paper makes marginal contributions (vs non-contemporaneous work), so I would rather not see it in the conference.

**Missing References:**

N/A

**Paper Topic And Main Contributions:**

This paper makes a thorough survey of existing transferability estimation methods and conducts an empirical study on the GLEU benchmark.

**Questions For The Authors:**

1. The conclusion should be revised. How to select a valid PLM for a target task efficiently should be clearly summarized in the conclusion based on your analysis.
2. The analysis does not have any problem, what do authors think about the novelty of this paper? It is an important factor to determine whether to receive this paper or not.


**Reasons To Accept:**

This paper lists existing transferability estimation methods and analyzes the performance of selected models. Experimental results show H-Score performs well with superiorities in effectiveness and efficiency.

**Reasons To Reject:**

This paper focuses on the analysis of existing transferability estimation methods, but the novelty is limited. Although as a survey paper, proposing a new method or evaluation metric is not necessary, the contribution of this paper should be expanded to be qualified to be published.

**Reproducibility:**

4: Could mostly reproduce the results, but there may be some variation because of sample variance or minor variations in their interpretation of the protocol or method.

**Reviewer Confidence:**

4: Quite sure. I tried to check the important points carefully. It's unlikely, though conceivable, that I missed something that should affect my ratings.

---

> ### Author Rebuttal · Authors · 2023-08-23
>
> Thanks for your comments!
> The novelty and contribution of this work are:
>        (1) summarizing as well as categorizing the previous works for model selection.
>        (2) providing a empirical study to compare current methods on GLUE benchmark.
> wihch are still lacking in the existing works.
>
> Answer to question#1:
> We will add the instructions to select PLM according to the empirical conclusion in the revised version.
>
> Answer to question#2:
> As a survey paper, the novelty of this work is to provide a taxonomy over model selection methods and also the empirical studies which have not been explored.

---

### Meta-Review · Area_Chair_143s · 2023-09-22

**Recommendation:** 3

**Metareview:**

The main contribution of this work is creating a taxonomy of methods to estimate the "transferability" of Pretrained Language Models (PLMs) based on whether they require certain inputs, are task-agnostic, take into consideration the dynamics of the features during fine-tuning, and/or are training-free. However, the properties of the surveyed methods are not always adequately expounded. Moreover, the Authors provide a comparison of their performance on tasks from GLUE. However, the selection of PLMs is limited to variants of BERT and RoBERTa encoders; hence, it is hard to determine if the findings generalise to auto-regressive and encoder-decoder PLMs. Finally, despite computer vision being prominent in the abstract and introduction, no studies have been carried out in this domain. Hence, there is a mismatch between the motivation and the experimental setup. Overall, this paper might be valuable as a summary of previous work on transferability, but its novelty is somewhat limited, as it mostly compares existing methods in a standard NLP benchmark. This submission might be considered for acceptance to Findings.

---

### Decision · Program_Chairs · 2023-10-07

**Decision:**

Accept-Findings

**Comment:**

The main contribution of this work is creating a taxonomy of methods to estimate the "transferability" of Pretrained Language Models (PLMs) based on whether they require certain inputs, are task-agnostic, take into consideration the dynamics of the features during fine-tuning, and/or are training-free. However, the properties of the surveyed methods are not always adequately expounded. Moreover, the Authors provide a comparison of their performance on tasks from GLUE. However, the selection of PLMs is limited to variants of BERT and RoBERTa encoders; hence, it is hard to determine if the findings generalise to auto-regressive and encoder-decoder PLMs. Finally, despite computer vision being prominent in the abstract and introduction, no studies have been carried out in this domain. Hence, there is a mismatch between the motivation and the experimental setup. Overall, this paper might be valuable as a summary of previous work on transferability, but its novelty is somewhat limited, as it mostly compares existing methods in a standard NLP benchmark. This submission might be considered for acceptance to Findings.